# ZCal: Calibrating radio interferometric data with machine learning

## Abstract

Calibration is the most critical data processing step needed for generating images of high dynamic range (CASA cookbook, 2009). With ever-increasing data volumes produced by modern radio telescopes (Aniyan & Thorat, 2017), astronomers are overwhelmed by the amount of data that needs to be manually processed and analyzed using limited computational resources (Yatawatta, 2020). Therefore, intelligent and automated systems are required to overcome these challenges. Traditionally, astronomers use a package such as Common Astronomy Software Applications (CASA) to compute the gain solutions based on regular observations of a known calibrator source (Thompson et al., 2017) (Abebe, 2015) (Grobler et al., 2016) (CASA cookbook, 2009). The traditional approach to calibration is iterative and time-consuming (Jajarmizadeh et al., 2017), thus, the proposal of machine learning techniques. The applications of machine learning have created an opportunity to deal with complex problems currently encountered in radio astronomy data processing (Aniyan & Thorat, 2017). In this work, we propose the use of supervised machine learning models to first generation calibration (1GC), using the KAT-7 telescope environmental and pointing sensor data recorded during observations. Applying machine learning to 1GC, as opposed to calculating the gain solutions in CASA, has shown evidence of reducing computation, as well as accurately predicting the 1GC gain solutions and antenna behaviour. These methods are computationally less expensive, however they have not fully learned to generalise in predicting accurate 1GC solutions by looking at environmental and pointing sensors. We use an ensemble multi-output regression models based on random forest, decision trees, extremely randomized trees and K-nearest neighbor algorithms. The average prediction error obtained during the testing of our models on testing data is $\approx 0.01 < rmse < 0.09$ for gain amplitude per antenna, and $0.2rad < rmse < 0.5rad$ for gain phase. This shows that the instrumental parameters used to train our model strongly correlate with gain amplitude effects than a phase.

## 1 Introduction

Modern-day astronomy is at an unprecedented stage, with a deluge of data from different telescopes. In contrast to conventional methods, today astronomical discoveries are data-driven. The upcoming Square Kilometer Array (SKA) is expected to produce terabytes of data every hour (The SKA telescope). With this exponential growth of data, challenges for data calibration, reduction, and analysis also increase (Aniyan & Thorat, 2017), making it difficult for astronomers to manually process and analyse the data (Yatawatta, 2020). Therefore, intelligent and automated systems are required to overcome these challenges. One of the main issues in radio astronomy is determining the quality of observational data. Astronomical signals are very weak by the time they reach the Earth's surface. They are easily corrupted by atmospheric interferences, incorrect observational parameters (e.g. telescope locations or telescope pointing parameters), malfunctioning signal receivers, interference from terrestrial man-made radio sources and tracking inaccuracies (Taylor et al., 1999). Therefore, it is required to do proper corrections to the observational data before processing the data. Radio astronomers spend a considerable amount of time performing a series of preprocessing steps called calibration, which involves the determination of a set of parameters to correct the received data. These generally include instrumental as well as astronomical parameters. The general strategy for

doing these corrections makes use of a calibrator source. Calibrator sources are well suited for determining astronomical parameters for data corrections because they have known characteristics such as the brightness, shape, and frequency spectrum (Taylor et al., 1999). This process of calibration is iterative and time-consuming. During scientific observations, different external parameters such as atmospheric pressure, temperature wind conditions, and relative humidity are collected through thousands of sensors attached to the telescopes and its adjoining instrumentation. The data coming from different sensors may provide information about the external conditions that may have corrupted the observed data. This piece of information is not always included in the conventional calibration steps. We propose to use machine learning methods to predict the calibration solutions, looking at pointing and environmental sensor data. This is mainly motivated by the fact that calibration steps make corrections to data that has been corrupted by environmental parameters. In this study, we make use of data from the Karoo Array Telescope (KAT-7), an array consisting of seven telescopes, which is a precursor to the MeerKAT radio telescope The SKA telescope. We look at eight types of sensor data recorded during observations, with a calibrator source PKS1613-586 to generate the training and testing dataset. The overall generated dataset contains sensor data per telescope and calibration solutions for the signal received by each telescope in horizontal polarization (H-pol) and vertical polarization (V-pol). These calibrator solutions are calculated using the astronomy software called Common Astronomy Software Applications.

## 2    CALIBRATION

In radio astronomy, ideally one might think that after obtaining the observed visibilities the next step would be to directly retrieve the actual visibilities of the target source and perform imaging. However, the measured visibilities $V^{obs}$ are different from the actual visibilities $V^{True}$ and this is due to instrumental and environmental effects (Richard Thompson et al., 2017). An example of these effects on the signal measured by a radio interferometry include antenna gains (slowly and fast time-varying instrumental part), atmospheric effects, pointing errors (tracking inaccuracies) and incorrect observation parameters (antenna pointing parameters). Signal effects are classified into two types, direction-independent effects (affecting the signal from all directions equally) and direction-dependent effects (which vary based on the sky position of the signal) (Taylor et al., 1999). These effects can be corrected by estimating the errors associated with the measured visibilities, thereby recovering the true visibilities. This process is called calibration. In its simplest form, calibration minimizes the error between observed and predicted (model) visibilities by estimating the correct complex instrumental gain response (Grobler et al., 2016). Suppose for baseline pair $(i, j)$, the observed visibility is $V_{i,j}^{obs}(t)$ and the true visibility is $V_{i,j}^{True}(t)$ at observation time t. The basic calibration formula is written as,

$$V_{i,j}^{obs} = G_{i,j}V_{i,j}^{True} + \epsilon_{i,j}(t) \tag{1}$$

where, $G_{i,j(t)}$ denotes the complex antenna gains for baseline $(i, j)$ as a result of unwanted effects and may vary with time (Thompson et al., 2001). The extra term $\epsilon_{i,j}(t)$ is a stochastic complex noise (Taylor et al., 1999). Most of the corruptions in data occur before the signal is correlated and the response associated with antenna $i$ does not depend on the response of antenna $j$. Note that the sources that are the subject of astronomical investigation will be referred to as "target sources" to distinguish them from calibrator sources (Thompson et al., 2001).

## 3    KAT-7 TELESCOPE

The KAT-7 is a seven-dish interferometry that was built as an engineering prototype for techniques and technologies in preparation for the 64-dish Karoo Array Telescope (MeerKAT) (Foley et al., 2016). These instruments are located in the Northern Cape Karoo desert region and are operated remotely from Cape Town. The construction of KAT-7 began in 2008 with the writing of the telescope requirements specification and was completed in 2010. It was then operated in engineering (commissioning) mode until its shut-down in 2016 (Foley et al., 2016).

### 3.1    SENSOR DATA

During science observations, different external parameters like atmospheric pressure, temperature wind conditions, and relative humidity are also collected through thousands of sensors attached

to the telescopes and its adjoining instrumentation. The data coming from different sensors may provide information about the external conditions that may have corrupted the observed data. This piece of information is not always included in the conventional calibration steps. We propose to use machine learning methods to predict the calibration solutions looking at pointing and environmental sensor data. This is mainly motivated by the fact that calibration steps do corrections to data that is corrupted by environmental parameters. In this study, we make use of the data from the Karoo Array Telescope (KAT-7). We look at pointing azimuth, elevation, scan, offset, temperature, wind speed, air pressure, relative humidity sensor data recorded during observations with a calibrator source $PKS1613 - 586$ to generate the training and testing dataset. The overall generated dataset contains sensor data per telescope and calibration solutions for correcting the signal received by each telescope in horizontal polarization (h-pol) and vertical polarization (v-pol). These calibrator solutions are calculated using one of the traditional astronomy software called CASA which is used for data calibration and imaging in radio astronomy.

## 3.2 Preparation of training data

The objective of this study is to find correlations between calibration solutions and sensor information on the telescope. Therefore, the main dataset for the study is the time-based sensor information of each antenna. The process of data collection encompasses all of the steps required to obtain the desired data in digital format. Methods of data collection include acquiring and archiving new observations, querying existing databases according to the science problem at hand, and performing as necessary any cross-matching or data combining (Ball & Brunner, 2010).

In every observation, the collected data are stored by the data capturing system in the Hierarchical Data Format (HDF5), which is a set of file formats designed to store and organize large amounts of data. The HDF5 file consists of two parts meta-data and observed visibilities. In meta-data one finds static information of the data set, including observer, dump rate and all the available subarrays and spectral windows in the data set selection criteria (antennas, channel frequencies, targets, scan information) and sensor data of interest as a function of time. The data observed by the radio telescope are in the form of complex numbers referred to as visibilities. Each source observed contains its own visibilities as a function of time along with sensor data, which keep a record of the telescope's activity and behaviour as these are observed.

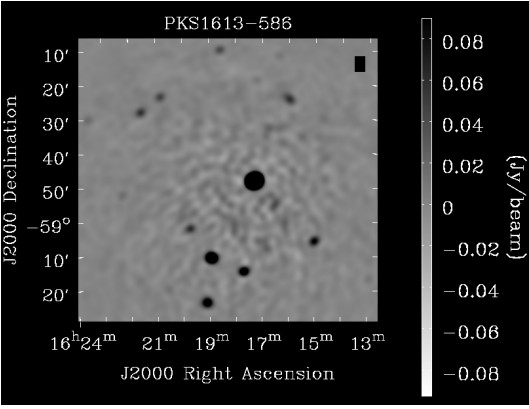

Figure 1: An image of the phase calibrator PKS1613-586 located at RA, Dec (J2000) using *CLEAN* task in CASA.

In preparation for the training and testing dataset, we look at environmental sensors and instrumental sensors recorded during observations with a flux calibrator and a phase calibrator source PKS1613-586 in Figure 1. The chosen sensors of interest from each observation are: air temperature, wind speed, wind direction, air pressure, relative humidity, actual refraction elevation, actual refraction azimuth, actual scan elevation, actual scan azimuth, actual pointing elevation and actual pointing azimuth.

## 4 PROPOSED METHOD

Different calibration techniques have been developed with the enhancement of the dynamics of the modern radio astronomy instruments to address these challenges raised by the new instruments, providing precise calibration performance. These techniques are loosely classified into first generation calibration (1GC), second generation calibration (2GC) and third generation calibration (3GC) (Noordam & Smirnov, 2010). In this study, we concentrate on generating 1GC calibration with the help of machine learning techniques. Our aim is to provide a machine learning model that predicts calibration solutions from sensor data from the telescope. This approach would help to speed up the calibration processes and decrease the time period of the calibrator monitoring, thus improving the time duration duration for tracking the target source observed as shown in 2.

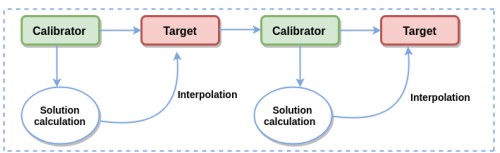

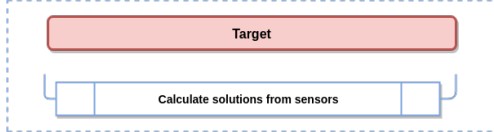

(a) Observation procedure and calibration using CASA

(b) Observation procedure using machine learning

Figure 2: Observation procedures

Several different approaches are employed in machine learning regression. These approaches learn the relationship between the input and output by fitting a model directly from the data. In this study, we consider tree-based approaches: Decision tree, random forest, extremely randomized tree, and the neighborhood search approach: (K-nearest neighbor) to tackle our problem. We call our approach ZCal model which is multi-output regression. We formulate our regression estimation problem as follows:

Suppose we have a feature matrix of sensor data

$$\mathbf{X}_t = \begin{bmatrix} x_{11} & x_{12} & x_{13} & \dots & x_{1n} \\ x_{21} & x_{22} & x_{23} & \dots & x_{2n} \\ \vdots & \vdots & \vdots & \ddots & \vdots \\ x_{d1} & x_{d2} & x_{d3} & \dots & x_{dn} \end{bmatrix} = (x_{i,j}) \in \mathbb{R}^{d \times n}, i \in \{1, 2, \dots, d\}, j \in \{1, 2, \dots, n\}$$

and corresponding complex target variables to learn and predict on,

$$\mathbf{Y}_t = \begin{bmatrix} y_{11} & y_{12} & y_{13} & \dots & y_{1m} \\ x_{21} & y_{22} & y_{23} & \dots & y_{2m} \\ \vdots & \vdots & \vdots & \ddots & \vdots \\ y_{d1} & y_{d2} & y_{d3} & \dots & y_{dm} \end{bmatrix} = (y_{k,l}) \in \mathbb{C}^{d \times m}, k \in \{1, 2, \dots, d\}, l \in \{1, 2, \dots, m\}$$

where each column represents a vector of length d, containing unique calibration solutions as function of time $t$ per observation represented as a complex variable $Ae^{i\phi} = A(\cos \phi + i \sin \phi)$ for each polarization $H\&V$ (Thompson et al., 2017).

Due to different physical causes on the received signal, we therefore choose to treat the antenna phases and amplitudes separately by splitting equation the complex variable into gain amplitude solutions $\left| Ae^{i\phi} \right|$ and gain phase solutions $\phi$.

We construct a learning machine, $M : \mathbf{X}_t \to \mathbf{Y}_t$, which when given a validation set of sensor examples, $\mathbf{X}_t^*$, minimises some measure of discrepancy between its prediction $M(\mathbf{X}_t^*) \approx \widehat{\mathbf{Y}}_t$, and the value of $\mathbf{Y}_t$, where $M$ represents the predictor. We measure the discrepancy using four commonly used statistical measures in regression (Borchani et al., 2015): coefficient of determination, explained variance, root mean squared error (RMSE) and root mean absolute error (RMAE).

The aim of this regression exercise is to predict multiple target variables $\widehat{\mathbf{Y}}_t$ hence it is referred to as multi-output regression. The learned model will then be used to predict multi-output values $\widehat{\mathbf{Y}}_{t+1}$

of all target variables of the new incoming unlabelled instances $\mathbf{X}_{t+1}$. It has been proven that multi-output regression methods provide means to model the multi-output datasets effectively and produce better predictive results (Borchani et al., 2015). This method does not only consider the underlying relationships between the features and the corresponding targets but also the relationships between the targets themselves, thereby producing simpler models with better computational efficiency (Borchani et al., 2015). Borchani et al. (2015) discuss several applications of multi-output regression including the challenges such as missing data, i.e., when some features or target variables are not observed.

## 5 RESULTS AND DISCUSSION

| Antenna | Extremely randomized phase | | | |
|---|---|---|---|---|
| | rmse | Rmae | R2score | Explained $\sigma^2$ |
| Uniform average-H | 0.337 | 0.336 | 0.936 | 0.936 |
| Uniform average-V | 0.235 | 0.301 | 0.956 | 0. 957 |
| | Extremely randomized tree amplitude | | | |
| Uniform average-H | 0.027 | 0.083 | 0.925 | 0.925 |
| Uniform average-V | 0.027 | 0.083 | 0.923 | 0.923 |

Table 1: The table shows the performance of the extremely randomized tree algorithm as the best method in predicting the amplitude and phase gain solutions for both H and V polarization as shown in figures 3. The values shown represents the uniform average of all KAT-7 antennas, i.e., all output measures are averaged with uniform weight. Most of the predictions stay near the ideal truth values with RMSE and RMAE $\approx< 0.5$, $R^2$ and explained variance $V$ are converging to 1.

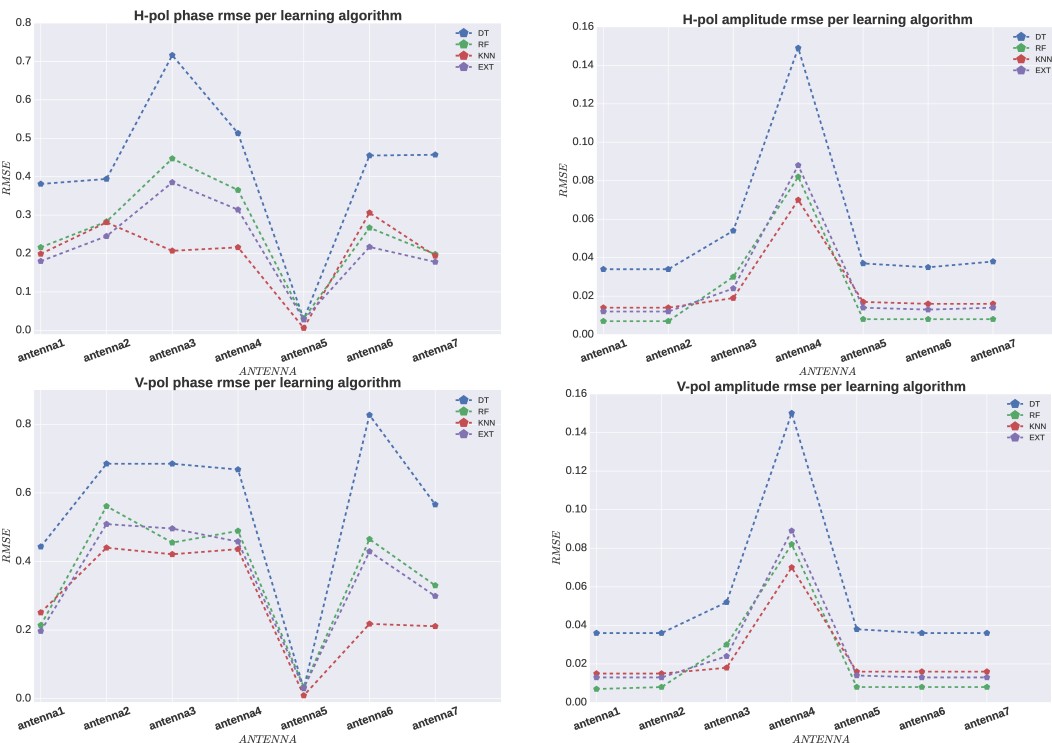

Figure 3: Root mean square error for each learning algorithm in predicting the H and V polarization phase and amplitude gain solutions for each antenna. The blue, green, red and purple lines represent the different learning algorithms used in the experiment.

From Figure 3, we observes that the learning algorithms have learned this behaviour pattern with rms error accuracy $\approx \leq 0.5$ Radians during test time, resulting in predicting zero phase gain solutions for

antenna 5 for both polarizations. Similarly for amplitude gain, we have achieved an error of less than 0.02 during training time. We validated our models looking at one observational data for a calibrator source PKS1613-586. We provide our models with sensor data during tracking of the calibrator source as input to our trained model. The phase and amplitude output predictions in Figures 4 and 5 shows that the models have learned to generalize for the reference antenna 5, by predicting zero phases for antenna 5 H and V polarization, assuming that the antenna was stable without any dropouts during the period of the observation with antenna 6 and antenna 7 being a bit nosier. In Figure 3, though the models were supposed to perform differently because of their parameter settings, one notices that the random forest, K-nearest neighbor and the extremely randomised trees methods are very close to one another in rms error as a function of antenna 1H, 2H, 5H, 6H and 7H, whereas there are large large rms error bars in each model per antenna 3H and 4H. Such behaviour gives us an idea about the instability of these two antennas. This is valuable information that will make contribution towards flagging corrupted data due to unstable antennas.

These models managed to learn the most critical part, i.e., the sinusoidal variation of the gain amplitude solutions over time as shown in Figures 5. However, We observe that the machine learning amplitude is performing lower than CASA solutions with a factor of 33.33%.

In Figure 5, The model predicts a drop in amplitude as a function of time in the middle of the observation from 0.12 to 0.09 level of CASA, while as the true amplitude have not dropped. From this unique behaviour observed, we can conclude that it is either due to prediction failure as the models have been trained on limited amount of observations and therefore failing on different observation settings, or it could be a behaviour that CASA did not detect as it does not include sensor data when calculating its calibration solutions.

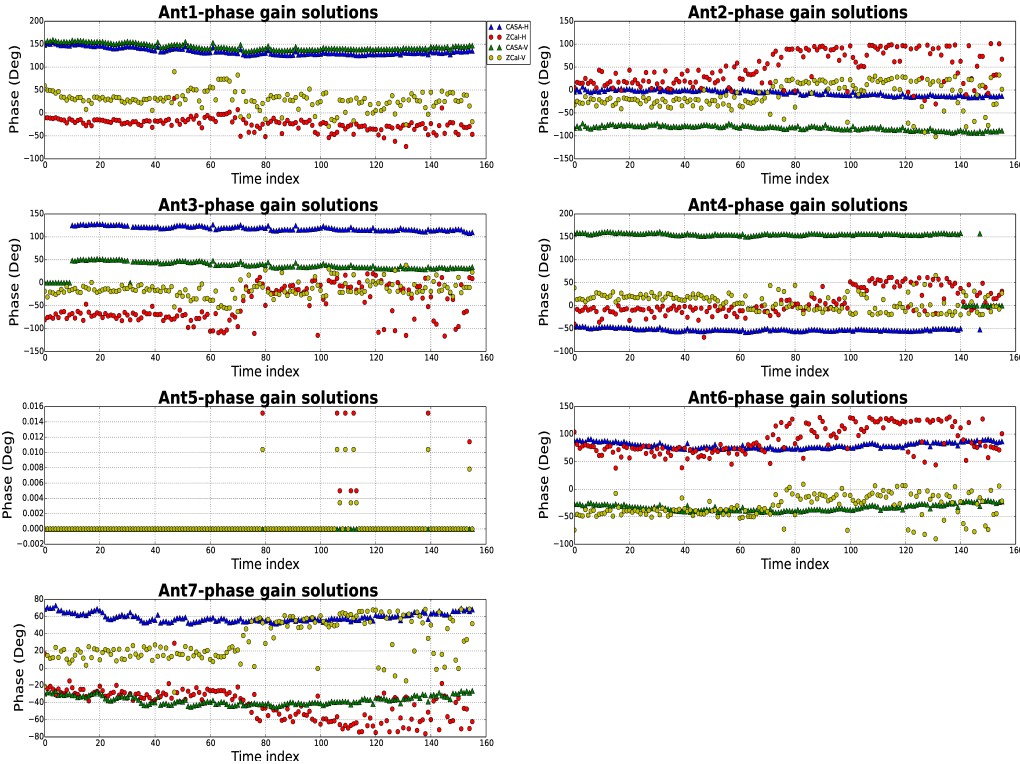

Figure 4: This figure shows the response of the extremely randomized tree learning algorithm in predicting the phase gain solutions for the calibrator PKS1613-586 from the validation data. In each subplot we compare the predicted phase gain solutions (where H polarization is represented by red and V-polarization is represented by yellow) with the CASA generated phase gain solutions (where H polarization is represented by blue and V-polarization is represented by green). The extremely randomized tree algorithm is failing to predict the gain phase solutions for some antennas and performing well in others with few outliers.

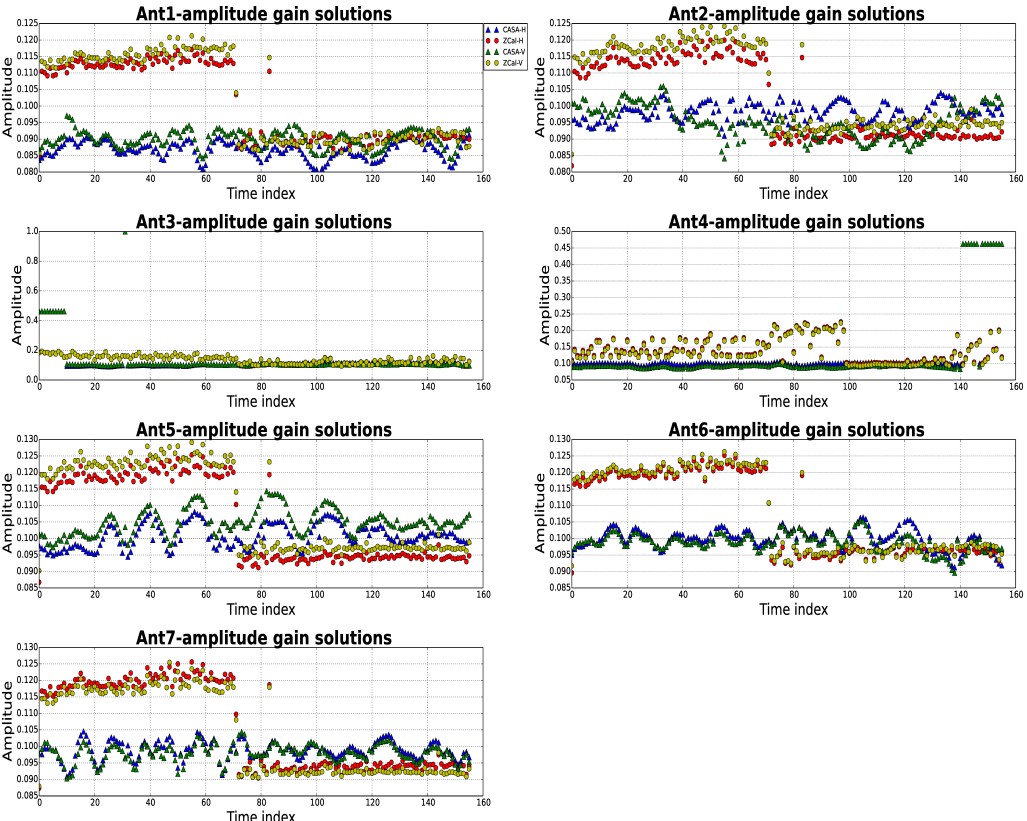

Figure 5: This figure shows the response of the extremely randomized tree learning algorithm in predicting the amplitude gain solutions for the calibrator PKS1613-586 from the validation data. In each subplot we compare the predicted amplitude gain solutions (where H polarization is represented by red and V-polarization is represented by yellow) with the CASA generated phase gain solutions (where H polarization is represented by blue and V-polarization is represented by green). The extremely randomized tree algorithm is failing to predict the gain amplitude solutions in all the antennas, for solutions at time index $< 70$. We further observe that at time index $80$ the algorithm performing better in predicting the amplitude solutions, slightly close to CASA on the new observation dataset.

## 6 CONCLUSION

We have shown that the application of machine learning techniques to telescope sensor data opens a new avenue in the development of calibration methods. We used the telescope sensor data to learn the variability of the complex gain calibration solutions for the calibrator PKS1613-586 as a function of time. The implementation of the ZCal algorithm is based on regression machine learning algorithms, to predict the calibration solutions and study each antenna's behaviour. Using the 1GC calibration solutions obtained with CASA, we constructed a matrix of training sample $n_L$ and testing sample $n_T$ to train the machine learning algorithms (decision tree, random forest, extremely randomised trees, and K-nearest neighbor) to be able to discern the patterns that relate complex gain solutions to external parameters. Since gain solutions are complex, we have implemented ZCal to learn on phase and amplitude separately. Each learning algorithm ran on the learning sample N times and its error was estimated on the test sample. We presented a statistical framework to measure the accuracy of each multi-output regression mode and our results are encouraging with an rms error of $\approx \leq 0.5\ rad$ during testing of our models using the testing data for gain amplitude and phase. Comparing the performance of these algorithms, the random forest, extremely randomized tree and K-nearest neighbours were shown to be the best for our purpose. We observed that the environmental and the pointing sensor readings were more strongly correlated to the amplitude than phase. Consequently, the ability to predict gain-phase was overall poor; gain-amplitude prediction

was accurate in some cases (capturing non-trivial behaviour such as oscillations), and completely failed in others. The purpose of the study was to show that machine learning techniques can make available connections "blindly" without access to physical intuition. The accurate prediction of gain-amplitudes, in some cases, suggests that this is indeed feasible. It is not clear what caused the failed predictions, although we can always speculate on physical differences between observations that our sensors were not sensitive to. We can therefore expect that with access to a larger array of sensors, the ZCal approach will be able to make better gain predictions.

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
