# OpenReview forum: "ZCal: Machine learning methods for calibrating radio interferometric data "
_ICLR.cc/2021/Conference — Reject_

### Official Review · AnonReviewer1 · 2020-10-14
**Investigation incomplete**

**Rating:** 4
**Confidence:** 5

**Review:**

In the manuscript, "ZCal: Calibrating radio interferometric data with machine learning", the authors consider the problem of calibrating radio telescope data against a source of known brightness, which they (correctly) identify as a multi-variate, multi-output regression problem.  Compared to a typical accepted ICLR paper the application area does not appear to be particularly novel with regard to methods deployed (off-the-shelf KNN, decision tree, random forests etc) or the size/structure of the dataset; and no new insights into these methods (such as might generalise to help solve other problems) are recovered.  For this reason I would be surprised if a revised draft could encourage me to change my recommendation to accept.  For what it's worth, I would have been interested to see a description of the CASA routine for calibration to understand the difference between this and the ML algorithms.

---

### Official Review · AnonReviewer3 · 2020-10-25
**No machine learning contribution**

**Rating:** 2
**Confidence:** 5

**Review:**

SUMMARY

This work proposes to use an ensemble of very well-known machine learning models (mainly tree-based methods) to calibrate radio interferometric data from the KAT-7 telescope. This provides a more efficient alternative to the traiditional approach, which is based on the astrophysicists workforce.

REASONS FOR SCORE

In my humble opinion, this work does not present a machine learning contribution of interest for the ICLR community. It applies standard and well-known approaches to an specific remote sensing problem.  I would point the authors to different venues, such as the IEEE International Geoscience and Remote Sensing Symposium.

Moreover, I would recommend the authors to compare their approach to some others methodologies used in the field. Right now, the experimental section only analyzes the results of the proposed approach, with no baselines.

---

### Official Review · AnonReviewer2 · 2020-11-01
**An interesting application, but not suited for this conference**

**Rating:** 3
**Confidence:** 4

**Review:**

# Summary
The paper presents a study of using machine learning methods to calibrate a radio telescope using information from sensor data on, e.g., atmospheric conditions. The authors consider tree- and neighbourhood-based methods for predicting amplitudes and phases for seven antennas. The results show that the methods perform quite well in terms of RMSE and explained variance.

# Evaluation
## Strong points
It is a timely and interesting application, and it is clear that the field of astronomy holds lots of potential for machine learning applications.

## Weak points
* From a machine learning point of view, the methods are very simple.
* The evaluation overall is confusing and somewhat weak. For instance, it is unclear to me whether the competing method, CASA, can be considered as the ground-truth or as a method to outperform.
* The evaluation based on figures 4 and 5 seem to be done qualitatively.

## Recommendation
Rejection.

The main reason for my recommendation is that ICLR is not the right venue for this paper. It does not deal with representation learning, and the methods are too basic for a top-tier machine learning conference in general.
In addition, the experimental evaluation, which is the main body of the paper, is quite weak and will need significant improvements.

## Detailed feedback
The methods considered in the paper (k-NN and tree-based methods) all base their predictions directly on the input features; thus, no representation learning is happening. Thus ICLR is not the right venue, and I doubt the paper would be accepted to a top-tier machine learning conference since it only considers (quite simple) off-the-shelf methods. KDD might be a better fit or an astronomical journal such as PASP.

Furthermore, there seems to be some confusion with regards to what a multi-output regression model is. As described in Borchani et al. (2015), multi-output regression methods also make use of correlations between outputs. It seems the authors are aware of this, as they cite the models' ability to make use of the 'relationships between the targets', which I assume refers to modelling correlations between the output dimensions. However, this is not how the methods used in the paper work. For both k-NN and tree-based methods, there is a clear separation between inputs and outputs, so the paper does not deal with multi-output regression either. This is not a problem per se, however, as it is not clear that a multi-output model would be better suited for the task considered here.

The paper completely lacks a section on prior work - both regarding related uses of machine learning and different ways to perform calibration.

Regarding the methods, and given that the predictions seem to be part of a time-series, I am not convinced the chosen ones are the most suited for the tasks. It would make sense to test methods from signal processing as well, for instance Gaussian processes or a recurrent neural network like an LSTM.
There are also no specific details on how the methods in the paper are constructed. Do they only use the instrumental and environmental sensor data as inputs, or do they use the temporal dimension in some way? Which hyperparameters were adjusted, and how were they selected?

There is very little information regarding the experimental setup. How large were the training and test sets? What are the dimensionalities of the inputs and outputs? It seems like only one source was considered; why is that? Only testing one source seems brittle. Similarly, was cross-validation used? If not, why? If it was, we need details on how it was set up and what it was used for (hyperparameter tuning, experimental evaluation, etc.).

For the experimental evaluation, I am confused about whether CASA is considered the ground-truth to compare with, or whether it is a method to outperform. Regardless, would it be possible to evaluate both CASA and your method on simulations to get an idea about how well each performs?

It also confuses me that antenna 5 is referred to as 'the reference antenna'. In what way was it used as a reference?

Further to the experimental evaluation, it would make sense to include the performance of a baseline or two, such as linear regression. I would also like to see error bars in figure 3, e.g., from cross-validation.

Since the paper deals with quite simple methods, I would expect some analysis of why they work and when they might fail. In figure 3, especially for the amplitude, some antennas have dramatically larger errors. What happens here? Also, which of inputs were most informative for predicting the phase and amplitude? You could get a feeling for the importances by, e.g., training the models on only one feature, or all features expect one, and repeating this for all features in turn.

# Questions
1. Why did you choose these particular machine learning methods?
2. How did you determine which method was the best? From the plots in figure 3, it looks like k-NN is slightly better overall.
3. Did you perform any cross-validation? If so, where are the error bars in figure 3? If not, why not?
  1. You mention 'large rms error bars in each model per antenna 3H and 4H'. Which error bars are you referring to?
4. Are the RMSEs in Figure 3 wrt. the CASA estimations?
  1. Also, to me (with no background in radio astronomy, though), it seems like a phase error of 0.3-0.4 is quite large (assuming one phase is 2\pi). How large errors can we expect from CASA?

# Suggestions for improvement
* Evaluate more than one calibration source. Without more sources, we can't be sure if your method performs consistently well.
* Figures 4 and 5: it might be useful to see instead residuals between your method and what you consider ground-truth.

---

### Decision · Program_Chairs · 2021-01-07
**Final Decision**

**Decision:**

Reject

**Comment:**

As discussed by several reviewers the paper is an application of classical ML approaches for a very relevant problem of calibration of radio interferometers. The application is interesting but lacks novelty in terms of ML methodology and the experiments do not provide a meaningful comparison between the state of the art and the proposed approach or justification for the choice of predictors and their parameters. This paper in clearly not a good fit for a general ML conference.